# IMPROVING HUMAN-AI COORDINATION THROUGH ONLINE ADVERSARIAL TRAINING AND GENERATIVE MODELS

**Paresh Chaudhary, Yancheng Liang, Daphne Chen, Simon S. Du, Natasha Jaques**
University of Washington, Seattle
`pareshrc, yancheng, daphc, ssdu, nj@cs.washington.edu`

## ABSTRACT

Being able to cooperate with diverse humans is an important component of many economically valuable AI tasks, from household robotics to autonomous driving. However, generalizing to novel humans requires training on data that captures the diversity of human behaviors. Adversarial training is a promising method that allows dynamic data generation and ensures that agents are robust. It creates a feedback loop where the agent's performance influences the generation of new adversarial data, which can be used immediately to train the agent. However, adversarial training is difficult to apply in a cooperative task; how can we train an adversarial cooperator? We propose a novel strategy that combines a pre-trained generative model to simulate valid cooperative agent policies with adversarial training to maximize regret. We call our method **GOAT**: **G**enerative **O**nline **A**dversarial **T**raining. In this framework, GOAT dynamically searches the latent space of the generative model for coordination strategies where the learning policy—the Cooperator agent—underperforms. GOAT enables better generalization by exposing the Cooperator to various challenging interaction scenarios. We maintain realistic coordination strategies by keeping the generative model frozen, thus avoiding adversarial exploitation. We evaluate GOAT with real human partners, and the results demonstrate state-of-the-art performance on the Overcooked benchmark, highlighting its effectiveness in generalizing to diverse human behaviors. [1]

## 1 INTRODUCTION

In multi-agent human-AI cooperation (Carroll et al., 2020), training a cooperative agent to generalize across diverse human behaviors remains a formidable challenge. Distribution shift, limited training data, and highly dynamic human decision-making make training these systems difficult. Without adequate coverage of all possible human interactions, AI agents learn brittle policies and struggle to generalize to novel, unseen behaviors in real human cooperation partners.

Adversarial training (Tucker et al., 2020; Cui et al., 2023; Fujimoto & Pedersen, 2022; Rutherford et al., 2024) could provide a computationally efficient solution for training robust Cooperators by exposing both the Cooperator agent's limitations and underexplored areas of the partner strategy space. It can be used to automatically generate a curriculum of challenging tasks targeted to the learner's weaknesses, resulting in a more robust policy Dennis et al. (2021). Yet despite its potential, adversarial training within cooperative AI introduces unique challenges; what does it mean to train an adversarial cooperator? In a single-agent task, Adversary's sole objective is to challenge the learning agent as much as possible by minimizing its performance. However, in a collaborative task, the adversarial agent has to find weaknesses in the learner while still simulating a realistic cooperative partner policy. If the adversary is not constrained to such policies, then it can become overly combative, resorting to strategies that sabotage the cooperation task, and is thus no longer a valid simulation of what a human cooperation partner might do. The challenge lies in simulating

---

[1] Please check out our **live interactive demo**, game play videos, and training code at our website `https://sites.google.com/view/goat-2025/home`

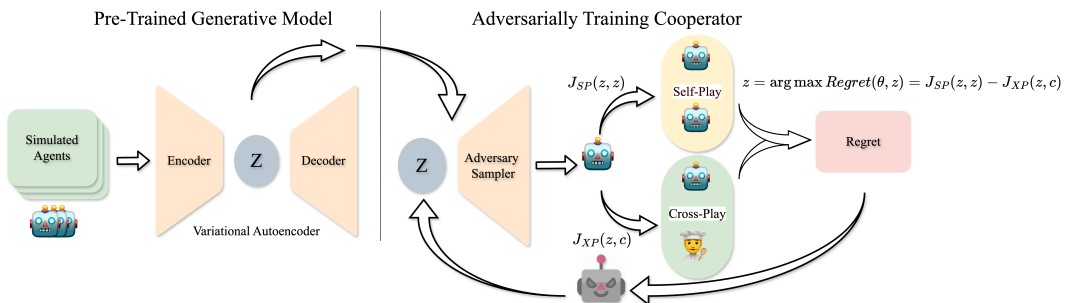

Figure 1: Adversarial training framework for cooperative agents. (Left) A generative model encodes simulated agents into a latent space to learn diverse agent strategies, which are then used to generate different types of training partners. (Right) GOAT samples new partners to maximize the Cooperator agent's regret, defined as the performance gap between self-play (the partner playing with itself) and cross-play (the partner playing with the Cooperator). The key idea is that the adversarial objective is constrained by the frozen generative model, which prevents it from generating self-sabotaging partners. But by applying regret-based adversarial training to search over the policies that can be generated by the model, we can expose the Cooperator to a curriculum of challenging training partners, ensuring it is robust to interacting with diverse human partners at test time.

*realistic* yet challenging partners that maintain cooperative intentions while simultaneously correcting weaknesses in the AI agent's policy.

For single-agent tasks, techniques like Unsupervised Environment Design (UED) (Dennis et al., 2021; Parker-Holder et al., 2022; Jiang et al., 2021) constrains the Adversary so that it can only generate valid tasks for which there is a solution. This is accomplished by maximizing the learning agent's *regret*, which is the performance gap between the optimal agent and the learning agent. The regret objective proves effective because it challenges the learning agent by continuously identifying difficult but achievable scenarios where the agent currently underperforms. In the cooperative setting, attempts have been made to create a regret-like objective (Charakorn et al., 2023; Villin et al., 2025; Rutherford et al., 2024; Wang et al., 2025; Erlebach & Cook, 2024). In methods like Charakorn et al. (2023), a diverse population is trained by optimizing for the difference between self-play and cross-play performance. However, naively applying **this type of adversarial objective can lead to policies learning to sabotage the game.** While attempts have been made to add additional objectives to counterbalance this effect (Sarkar et al., 2023). However, this approach is not able to address the issue that minimizing cross-play inherently incentivizes overly adversarial, uncooperative behavior.

How can we ensure that the adversarial agent can only generate **cooperative behaviors** that do not engage in self-sabotage but can still fully explore the space of challenging partner strategies? We propose a novel adversarial training method to achieve these criteria. We first pre-train a generative model of partner strategies on diverse cooperation data, following Liang et al. (2024). We then place this generative model into an adversarial training loop, where we train an Adversary to search for embedding vectors that, when passed through the generative model, decode into simulated partners that maximize the Cooperator's regret. Because we only update the weights of the Adversary and not the generative model, we ensure that all generated partners are cooperative, thereby **addressing the intentional sabotage seen in adversarial training.** In GOAT, the Adversary is motivated to maximize the regret, searching for valid coordination strategies where the Cooperator agent underperforms. In contrast, the Cooperator attempts to adapt to the Adversary and minimize the regret. Unlike previous adversarial approaches that risk learning degenerate strategies, this cyclic training method provides a natural curriculum for the Cooperator agent and unlocks the benefits of adversarial training for robustly generalizing to cooperate with diverse partners.

We evaluate GOAT on three distinct domains: 1) One-Step Cooperative Matrix Game (CMG); 2) Cooperative Reaching Game (CRG); and 3) Overcooked (Carroll et al., 2020), a popular human-AI zero-shot coordination benchmark. We compare to 5 competitive baselines (Zhao et al., 2022; Sarkar et al., 2023; Liang et al., 2024; Strouse et al., 2022; Carroll et al., 2020) and show that GOAT achieves improved performance across all three popular environments. In the Overcooked environment, we tested GOAT live against real, novel human partners and found that it significantly improves cooperation performance.

The contributions of this paper are as follows:

- We introduce **a novel adversarial training method for zero-shot coordination that trains both the Adversary and Cooperator agents online in a common payoff game**, resulting in a **dynamic curriculum** where the Adversary searches over the space of partner policies that a generative model can simulate. The generative model is limited to simulating cooperative partners, which ensures the adversarial training process generates challenging but realistic cooperative partners.

- We outperform 5 competitive baselines across 3 distinct popular cooperative environments.

- We perform a **real-time evaluation with novel human partners**, using the popular Overcooked benchmark (Carroll et al., 2020) commonly used in human-AI cooperation research like Strouse et al. (2022); Charakorn et al. (2023); Zhao et al. (2022); Sarkar et al. (2023); Liang et al. (2024). As compared to recent high-scoring techniques, we show that our method leads to the best cooperation performance with diverse people.

## 2 RELATED WORK

**Zero-shot coordination (ZSC).** Human-AI coordination requires training cooperative agents that can generalize to interactions with different humans in zero-shot settings, and thus navigate a vast coordination behavior space. (Hu et al., 2020; Kirk et al., 2023). Behavior cloning (Carroll et al., 2020) is limited by expensive and often scarce human data. Self-Play (SP) (Samuel, 1959; Silver et al., 2017; 2018; OpenAI et al., 2019) is an automatic training algorithm that does not need human data but converges to rigid policies that generalize poorly to novel human partners. Population-Based Training (PBT) (Jaderberg et al., 2017; Vinyals et al., 2019; Parker-Holder et al., 2020; Jung et al., 2020) has become a popular automatic training paradigm for ZSC, in which a Cooperator policy is trained using a large population of simulated partners (Hong et al., 2018; Strouse et al., 2022; Lupu et al., 2021; Charakorn et al., 2023; Zhao et al., 2022; Sarkar et al., 2023).

**Population diversity objectives.** In an effort to ensure the simulated agent population covers diverse human strategies, various methods (Hong et al., 2018; Parker-Holder et al., 2020; Wu et al., 2023; Lupu et al., 2021; Zhao et al., 2022; Charakorn et al., 2023; Sarkar et al., 2023; Rahman et al., 2023; Yuan et al., 2023) improve PBT by optimizing the diversity of the agents in the population. The population can be created by reward-shaping (Tang et al., 2021; Yu et al., 2023), manual design (Ghosh et al., 2020; Wang et al., 2022; Xie et al., 2021), and quality diversity (Canaan et al., 2019; Pugh et al., 2016; Wu et al., 2023; Fontaine & Nikolaidis, 2021). However, these methods typically use domain knowledge and fail to generalize and scale. Statistical methods like (Hong et al., 2018; Eysenbach et al., 2018; Parker-Holder et al., 2020; Derek & Isola, 2021; Lupu et al., 2021; Zhao et al., 2022) automate the diverse population generation, but Sarkar et al. (2023) demonstrated that trajectory distribution methods can result in minor variations in the policies rather than behaviorally different policies.

**Min-Max optimization.** Another line of work maximizes diversity by training self-play policies while minimizing cross-play rewards with the policies in the population (Cui et al., 2023; Charakorn et al., 2023; Rahman et al., 2023; 2024; Villin et al., 2025; Sarkar et al., 2023). However, agents might learn to sabotage the cross-play game while retaining high self-play performance with *handshaking* protocols, such that if their partner fails to complete the handshake correctly, they throw the game. To address this, Sarkar et al. (2023) introduces mixed-play, balancing self-play and cross-play by randomly sampling initial game states from both. However, such a solution still leads to rigid policies that fail to generalize, and does not address the fundamental problem that training a partner to minimize cross-play incentives sabotage. In this work, we only sample partners from a frozen, pre-trained generative model; since we do not train the partner policy on the adversarial objective, we prevent handshaking and sabotage. In addition, unlike prior work on this topic, we use online adversarial training to find and exploit weaknesses in the cooperator, instead of using adversarial objectives to pre-train a diverse partner population.

**Generative models for Cooperation.** Derek & Isola (2021); Wang et al. (2023); Liang et al. (2024) show that generative models can be trained on policies to simulate diverse and valid collaborative policies. Recently, Liang et al. (2024) proposed training a Variational Autoencoder (Kingma & Welling, 2022), on simulated cooperative agent trajectories to generate synthetic training data to improve human-AI coordination. This approach outperforms previous methods because the generative model enables sampling random novel partner strategies that interpolate between or combine strategies

from the data. GOAT leverages this approach to proactively generate challenging novel partners strategies to train the cooperator robustly.

# 3    PRELIMINARIES

**Two-player Markov game:** We consider the two-player Markov game (Littman, 1994) as a tuple $\left(\mathcal{S}, \mathcal{A}_1, \mathcal{A}_2, r, \mathcal{T}, \gamma\right)$ where $\mathcal{S}$ is the state space shared by all the agents. $\mathcal{A}_1$ and $\mathcal{A}_2$ denote the action space of agents such that at each time step in the environment, and each agent selects one action $a_{1,t} \in \mathcal{A}_1$, $a_{2,t} \in \mathcal{A}_2$ sampled from their own action space according to their policies $\pi_1 : \mathcal{S} \to \mathcal{P}(\mathcal{A}_1)$ and $\pi_2 : \mathcal{S} \to \mathcal{P}(\mathcal{A}_2)$. After taking these actions, they receive a shared immediate reward $r : \mathcal{S} \times \mathcal{A}_1 \times \mathcal{A}_2 \to \mathbb{R}$. The system then transitions to the next state $s_{t+1}$ according to the transition dynamics $\mathcal{T}(s_{t+1} \mid s_t, a_{1,t}, a_{2,t}) \in \mathcal{P}(\mathcal{S})$ where $\mathcal{P}(\mathcal{S})$ denotes the set of probability distributions over $\mathcal{S}$. The parameter $\gamma \in [0, 1)$ is the discount factor that weights immediate versus future rewards. The expected joint return is captured by $J(\pi_1, \pi_2) = \mathbb{E}\left[\sum_{t=0}^{\infty} \gamma^t r(s_t, a_{1,t}, a_{2,t})\right]$ which sums the discounted rewards from each timestep.

**Generative model of cooperation strategies:** Following Liang et al. (2024), we train a Variational Auto-Encoder (VAE) (Kingma & Welling, 2022) on diverse cooperative strategies. The VAE is trained to reconstruct the partner's actions from a dataset of coordination trajectories $D$, which is obtained from interactions between simulated PBT (Population-Based Training) agents within the environment. Here, we use a fixed agent population with training methods like Maximum Entropy Population (Zhao et al., 2022) and CoMeDi (Sarkar et al., 2023) to provide the simulated partner agents. The VAE consists of an encoder $q(z|\tau; \phi)$, which maps a trajectory $\tau = (s_0, a_0, ..., s_T)$ to a latent space, where $\phi$ represents the parameters of the decoder. The decoder $p(a_t|z, \tau_{0..t-1}; \phi)$ predicts actions $a_t$ autoregressively using the latent $z$ and the agent's history $\tau_{0..t-1}$. The VAE model is optimized using the ELBO loss described in (Kingma & Welling, 2022). The distribution of partner policies is therefore learned through the encoder's posterior $q(z|\tau)$ over the dataset, which maps trajectories with different cooperation styles to different regions of the latent space. The KL term regularizes these posteriors towards the prior $p(z) = \mathcal{N}(0, I)$. Sampling $z \sim \mathcal{N}(0, I)$, the decoder $p(a_t|z, \tau_{0..t-1}; \phi)$ generates a partner policy corresponding to one cooperation style, and diverse policies can be generated by sampling multiple latents from the prior. VAE thus learns a smooth distribution over partner policies in dataset $D$. The $\beta$ parameter that balances reconstruction accuracy against the KL divergence in the ELBO loss, ensuring the latent space resembles a standard normal distribution $\mathcal{N}(0, I)$. The choice of $\beta$ leads to a tradeoff between reconstruction quality and encoding disentangled latent representations (Burgess et al., 2018). We experiment with various $\beta$ values to get better disentangled representations.

# 4    GOAT: GENERATIVE ONLINE ADVERSARIAL TRAINING

We introduce GOAT, which combines generative modeling with online regret-based adversarial training to enable robust human-AI coordination.

**Adversarially minimizing Cross-Play (XP) performance.** We begin by introducing an adversarial training objective based on the XP performance of the Cooperator $\pi_C$ and an adversarial partner $\pi_A$. The adversary $\pi_A$ attempts to minimize cooperation performance, while a Cooperator policy $\pi_C$ attempts to maximize it. This type of optimization is widely known as minimax optimization in game theory:

$$J_{XP} = \max_{\pi_C} \min_{\pi_A} J(\pi_C, \pi_A) \tag{1}$$

As may be obvious, attempting to train agents to cooperate via XP minimization leads to a fundamentally misaligned optimization objective because it conflicts with the cooperative objective of maximizing common payoff. The Adversary agent is incentivized to develop behaviors that intentionally sabotage interactions rather than generate meaningful cooperation trajectories.

**Generating adversarial cooperative agents.** To address the above issues, we propose adding a pre-trained generative model (VAE) with frozen weights into the adversarial training loop. We hypothesize that adding a generative model trained solely on cooperative policy data will ensure that adversarial training generates only valid cooperative partners. By design, the generative model is not

capable of simulating self-sabotage or handshaking behaviors. Instead, we train a separate Adversary policy to search for embedding vectors $z$ that decode into partners that minimize XP performance for the specific Cooperator agent we seek to train. The Adversary can thus search over a well-structured, continuous latent space, allowing a smooth optimization landscape.

We formalize this adversarial training algorithm by modifying the previous $J_{XP}$ formulation in Eq. 1. Now, the Adversary policy conditions on a randomly sampled $z \sim \mathcal{N}(0, I)$, and transforms it to produce an Adversary embedding $z' = \pi_A(z)$. This modified $z'$ is fed to the decoder $p(a_t | z', \tau_{0:t-1}; \theta)$ of the generative model to produce the actions of the simulated partner policy $\pi_P$. Let $\pi_P^{\pi_A(z)}$ be the partner policy produced by having the adversary $\pi_A$ choose an embedding which is then simulated with the generative model. Then, the optimization problem is as follows:

$$\max_{\pi_C} \min_{\pi_A} \mathbb{E}_{z \sim \mathcal{N}(0,I)} \left[ \sum_{t=0}^{\infty} \gamma^t \, r(s_t, \pi_C(a_t^C | \tau_{0:t-1}), p(a_t^P | \pi_A(z), \tau_{0:t-1}; \theta)) \right] \quad (2)$$

$$= \max_{\pi_C} \min_{\pi_A} \mathbb{E}_{z \sim \mathcal{N}(0,I)} [J(\pi_C, \pi_P^{\pi_A(z)})] \quad (3)$$

This objective no longer directly trains the partner policy on the adversarial objective, but instead uses the adversary to steer the pre-trained generative model to select partners.

**Regret-based adversarial cooperative training.** Because Eq. 3 is still fundamentally a minimax objective, it does not guarantee effective exploration in the partner space because it optimizes for worst-case returns. Therefore, the Adversary will only search for the worst agents that the generative model can simulate. Instead, we would like to formulate a better objective that encourages searching for meaningful partner policies that could have good performance, but for which the Cooperator still fails to perform well. Thus, we propose using a regret objective to generate a realistic curriculum for the Cooperator agent. As proposed initially by Robbins (1952), regret is the performance gap between the returns of an agent-chosen strategy and the returns of the optimal strategy that could have been selected with perfect foresight. We formalize regret in the cooperative setting as the performance gap between the cross-play (XP) performance an agent achieves with a partner $\pi_P$, and $\pi_P$'s optimal self-play (SP) performance, $\mathbb{E}[J(\pi_P, \pi_P)]$, which is the score it achieves when paired with itself.

$$Regret(\pi_P, \pi_C) = \mathbb{E}[J(\pi_P, \pi_P) - J(\pi_P, \pi_C)] = J_{SP} - J_{XP} \quad (4)$$

**GOAT objective.** At the start of the training process, GOAT samples $z$ from the standard normal distribution and transforms it to the effective latent vector $z'$. This latent vector $z'$ is then used to generate the partner policy $\pi_P$ using the frozen weights of the VAE decoder $p(a_t | z', \tau_{0:t-1}; \theta)$. The partner policy $\pi_P$ is then paired with itself to evaluate SP returns and with $\pi_C$ to evaluate XP returns. Regret is evaluated after collecting the returns from both games. The Adversary is trained to choose $z'$ to maximize regret, while the Cooperator policy is trained to maximize its cooperation performance. The full objective of the GOAT game is thus:

$$\min_{\pi_C} \max_{\pi_A} Regret(\pi_P^{\pi_A(z)}, \pi_C) \quad (5)$$

The online optimization, as shown in algorithm 1, ensures that the Adversary is incentivized to continually discover new partner strategies that are challenging for the Cooperator, by searching the latent space to maximize regret. The Cooperator, in turn, learns to adapt to these partner strategies by minimizing the regret. This cyclic behavior naturally trains the Cooperator to learn to cooperate with diverse strategies, from more straightforward to increasingly complex ones. For invalid partners, the optimal policy cannot achieve positive returns, resulting in zero regret and providing no incentive to the Adversary. This property of regret promotes exploration that leads to curriculum generation to increasingly generate complex yet solvable tasks to train the Cooperator agent.

## 5 EXPERIMENTS

Our experimental evaluation is designed to assess how well our adversarial training procedure can improve zero-shot cooperation performance, both in simulation and with real humans.

**Environments.** We evaluate GOAT using three popular coordination benchmarks: 1) One-Step Cooperative Matrix Game (CMG) (Lupu et al., 2021; Zhao et al., 2022; Charakorn et al., 2023), 2)

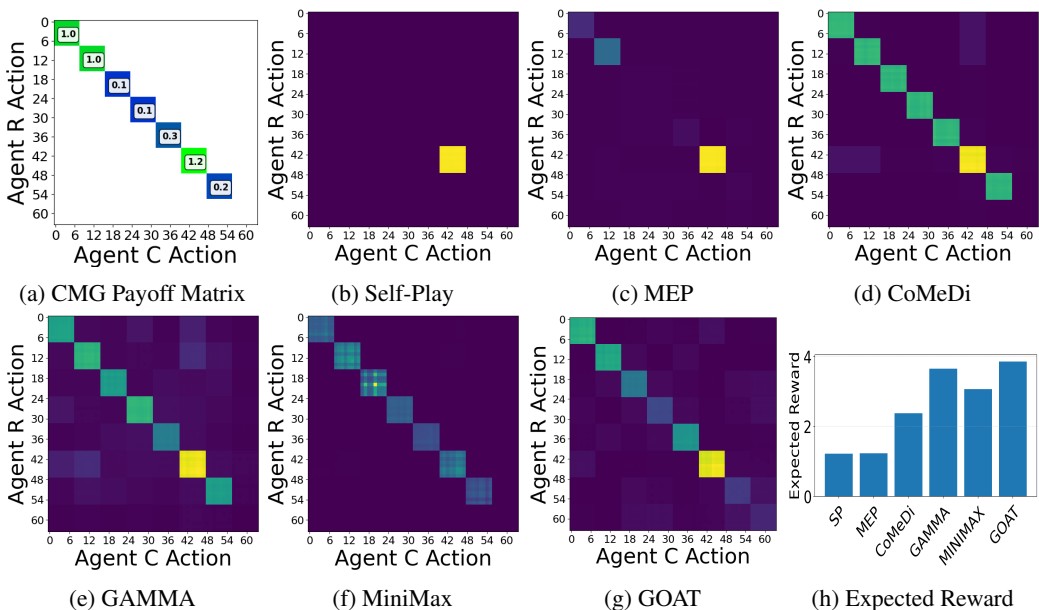

Figure 2: $a$) Cooperative Matrix Game. $b$) to $g$) Policy probability distribution to show coverage of different methods on the CMG payoff matrix $h$) Total expected rewards for each method, assuming we uniformly sample partners and each method gets a payoff for each partner proportional to the amount of coverage they have for the reward block.

Cooperative Reaching Game (CRG) (Rahman et al., 2023; 2024; Charakorn et al., 2023), and 3) Overcooked (Carroll et al., 2020).

- **One-Step Cooperative Matrix Game (CMG).** CMG is a single-step game where two agents (Row and Column) independently select an action by choosing a row and a column, respectively, and the intersection of their choices returns the associated reward according to the payoff matrix. Both agents receive the same reward. A typical payoff matrix in CMG is defined by $(M, \{k_m\}, \{r_m\})$ where $M$ is the number of solutions, $k_m$ is the number of compatible actions, and $r_m$ is the reward associated with each solution. We use a challenging layout shown in figure 2a with three scenarios: no reward, suboptimal rewards, and global maximum reward.

- **Cooperative Reaching Game (CRG).** CRG is a 5x5 grid world, with $|\mathcal{A}| = 5$, where two agents need to reach the same goal location simultaneously to receive rewards. For agents to cooperate and receive rewards, they must reach and stay in any of the same goal coordinates. An ideal Cooperator should chase their partner to the same goal or persuade them to follow the same goal. We evaluate GOAT and prior baselines against 11 heuristic agent teammates described in detail in appendix E.

- **Overcooked.** A state-of-the-art collaborative, real-time benchmark where two players need to cooperate by dividing cooking tasks and avoiding obstructing each other. Identifying the opposite player's intent and behavior is key to determining how to coordinate with the other agent

---

**Algorithm 1** GOAT

> Training VAE on trajectory dataset $D = {\tau_i}_{i=1}^N$ where $\tau = \{s_i, a_i, r_i\}_{i=1}^N$
> **Initialize:** VAE Decoder $p(a_t|z', \tau_{0..t-1}; \theta)$, Adversary $\pi_A$, Cooperator $\pi_C$
> **while** not converged **do**
>      Sample $z \sim \mathcal{N}(0, I)$
>      Map $z' = \pi_A(z)$
>      Generate $\pi_P = p(a_t|z', \tau_{0..t-1}; \theta)$
>      Collect Partner self-play returns $J(\pi_P, \pi_P)$
>      Collect Partner-Cooperator returns $J(\pi_P, \pi_C)$
>      Compute REGRET $= J(\pi_P, \pi_P) - J(\pi_P, \pi_C)$
>      Train Adversary Policy ($\pi_A$) with RL to maximize REGRET
>      Train Cooperator Policy ($\pi_C$) with RL to minimize REGRET
> **end while**

---

effectively. We test GOAT on the most challenging tasks with a diverse strategy space. Counter Circuit introduced in Carroll et al. (2020), and a more complex layout introduced by Liang et al. (2024), Multi-Strategy Counter.

**GOAT implementation & Baselines:** Training the Adversary is RL-algorithm-agnostic as it does not depend on state and is a one-step optimization process. To keep the optimization loop simpler, the Cooperator is trained using PPO, and the Adversary is trained using REINFORCE. Details on the KL-regularization of the VAE, training procedure for GOAT, and other architecture and hyperparameters are available in the Appendix. We compare to 5 competitive prior baselines on human-AI coordination: Behavior Cloning (BC) partners with RL cooperator (Carroll et al., 2020) (BC+RL), PBT methods Fictitious Co-Play (FCP) (Strouse et al., 2022), maximum entropy population (MEP) (Zhao et al., 2022), CoMeDi Sarkar et al. (2023), and GAMMA trained on MEP (MEP+GAMMA) Liang et al. (2024), which previously showed state-of-the-art performance in real human evaluations.

**Human Evaluation:** We conducted a user study with 40 participants to discover which method best coordinated with humans in a real-time evaluation in Overcooked, recording the team scores. Participants were recruited from the Prolific online crowdsourcing platform following an IRB-approved protocol. Each participant was tasked with completing 6 total rounds of the Overcooked game with an AI agent partner, where each round presented a different agent model among one of the baselines or our method. In order to minimize trends due to gameplay order, the agent models are loaded in randomized order, with a randomly sampled checkpoint chosen from 1 of 5 random seeds.

# 6 EXPERIMENTAL RESULTS

**RQ1: Can GOAT discover and exploit multiple coopera-tive strategies?** For GOAT to be effective at generalizing to novel partners, the adversary should be able to discover and exploit all rewarding cooperative modes. We first test this question with the CMG environment, using a population size of 8 for all baseline PBT methods. To train the VAE using the population of CoMeDi agents, we use the same procedure as GAMMA Liang et al. (2024), which pairs each of the 8 row and column agents in the population together to generate trajectory data. Since VAE does not have a population, it can generate many more agents than PBT methods.

As seen in Figure 2b and 2c, the baselines' Self-Play and MEP fail to cover all modes, implying they would not generalize to partners that used those modes. CoMedi 2d and GAMMA 2e discover all the regions and give the highest priority to the global maximum, but spread probability mass roughly equally across all other modes, regardless of their payoff. Whereas MiniMax 2f converges to the worst-case scenario. In contrast, GOAT not only covers all the modes but also is better at assigning higher probability to more rewarding modes (for example, assigning low probability to modes 3, 4, and 7, which have low payoff). Thus, GOAT achieves good coverage while also focusing on maximizing return. This is also evident in Figure 3, which shows the results of evaluating each method in the CRG environment. Here we see that GOAT achieves the highest average score when teamed with all 11 heuristic partner agents.

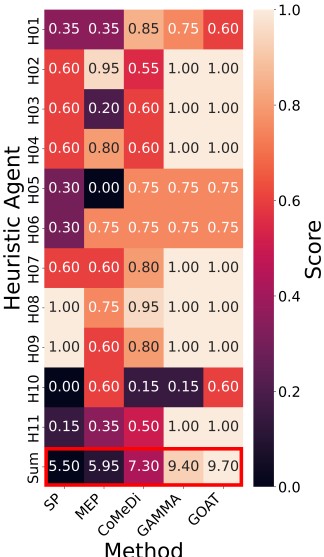

Figure 3: CRG: Average reward obtained against the 11 Heuristic Agent teammates across 5 seeds of each method. Sum row highlighted in red shows the summation of rewards, where the maximum possible reward is 11.

**RQ2: Will adversarial training result in a more robust Cooperator with better sample complexity in simulation?** PBT methods rely on simulated populations of self-play partners, and have poor coverage of actual human behaviors. Training a Cooperator against them may not achieve satisfactory performance with people, and it may take a long time to converge since the training does not focus on correcting weaknesses in the Cooperator, but instead randomly samples from the population. We hypothesize that our adversarial training technique can more efficiently search the space of partners to make an agent that is more robust and

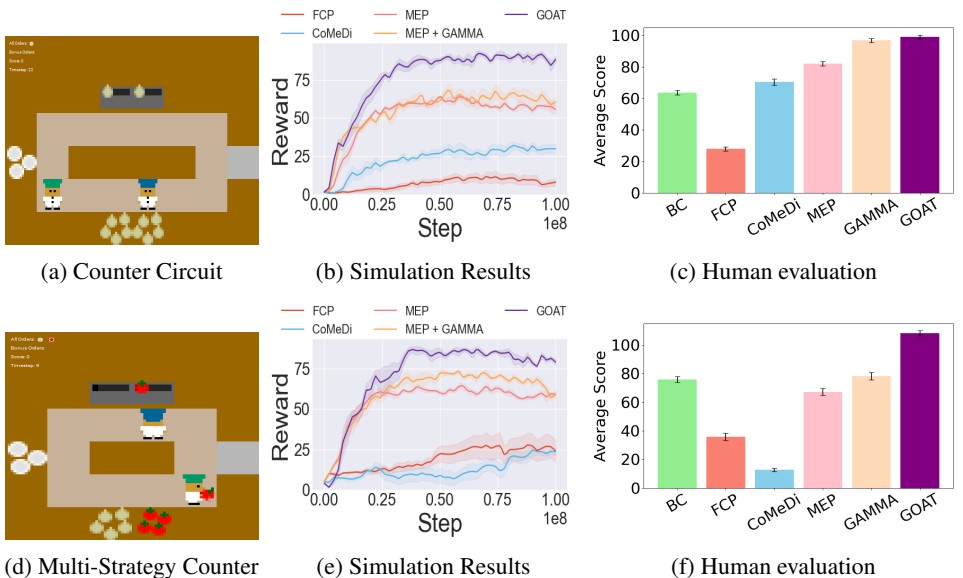

Figure 4: Overcooked: Carroll et al. (2020) first introduced the most challenging *Counter Circuit layout* (*a*). To increase coordination complexity, Liang et al. (2024) introduced the *Multi-Strategy Counter layout* (*d*), where players could choose between tomato and onion ingredients to cook soup. This adds an additional coordination challenge where agents have to adapt to soup-making strategies based on different ingredients. (*b*) & (*e*) are evaluation of GOAT against a human proxy model trained with BC. We compare to 4 baselines, which are trained using simulated population data, including the previous state-of-the-art method, GAMMA (Liang et al., 2024). Error bars show the std. err. over 5 random seeds. (*c*) & (*f*) shows the evaluation of the performance of methods when tested against real humans in two layouts: counter circuit and multi-strategy counter, respectively.

requires less training time. To validate this hypothesis, we investigate the Overcooked learning curves of all simulated training methods, including our own, in Figure 4b and 4e. Here, we evaluate the cooperation performance of the agents against a Human Proxy model (behavior-cloned (BC) model trained on human data) as in prior work (Carroll et al., 2020; Strouse et al., 2022; Liang et al., 2024). The human proxy enables the automatic evaluation of agents' performance throughout training.

As is evident in the curves, GOAT achieves significantly higher performance than all 4 baselines in both Overcooked layouts and increases performance more quickly than other techniques. We hypothesize that adversarial exploration in the generative model's latent space enables rapid coverage of diverse human behaviors, ranging from simple coordination strategies to complex, context-dependent strategies. By constraining the Adversary to valid cooperative policies, the method avoids degenerate strategies and exposes the Cooperator to all the semantically diverse strategies that are beneficial for generalizing to diverse partners. It efficiently targets training to weak points in the Cooperator's policy, and thus has lower sample complexity than traditional PBT methods.

**RQ3: Can GOAT achieve improved cooperation performance when evaluated in real-time with novel humans?** The real test of a human-AI coordination algorithm is whether it can work with actual people. As described in Section 5, we evaluated the Cooperator agents trained with GOAT and the 5 baselines alongside real human partners in the Counter Circuit and the Multi-Strategy Counter Overcooked layouts. We recruited novel human players with no prior exposure to the agents. Figure 4c and 4f depict the evaluation results for the two layouts and all the agents against human players. GOAT outperforms the previous baselines on both layouts. Even though GOAT shows significant improvement over baselines on Counter Circuit simulation results, the magnitude of the improvement over GAMMA is smaller (3%) on this layout. This is because both methods are reaching close to optimal performance in this simpler environment. However, on the more complex layout, Multi-Strategy Counter, GOAT leads to markedly higher cooperation performance with real humans, with a 38% improvement over the previous state of the art. We encourage readers to test these differences for themselves by visiting our webpage, which provides an interactive demo in which you can play Overcooked with both GOAT and the baselines https://sites.google.com/view/goat-2025/home.

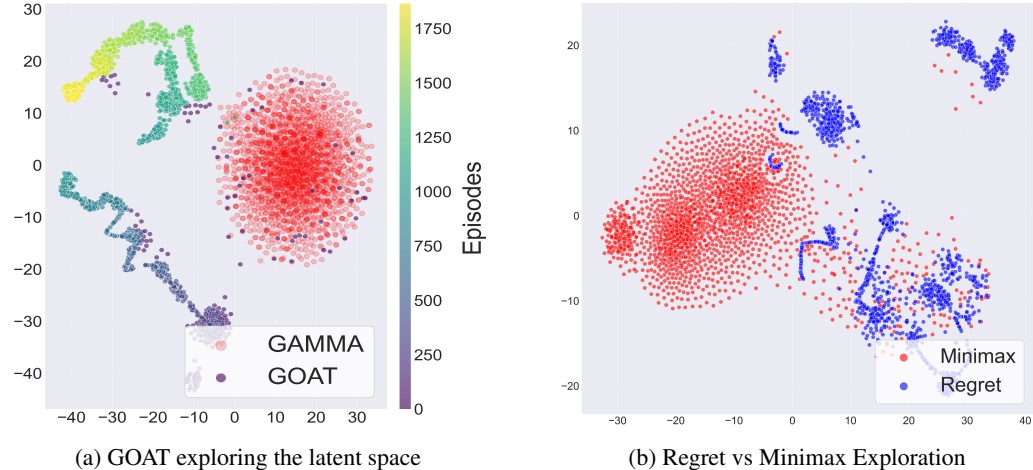

(a) GOAT exploring the latent space  (b) Regret vs Minimax Exploration

Figure 5: $a$) GOAT explores the latent space of the generative model across training episodes, progressing from episode 0 (purple points) to episode 1750 (yellow points). At the same time, GAMMA (red) remains stationary, maintaining a standard normal distribution cluster. $b$) Minimax (red) mostly fixates on regions of low reward or poor behavior, while Regret (blue) explores more broadly and generates a variety of challenging training scenarios.

**RQ4: Will adversarial training create a curriculum of increasingly challenging strategies by discovering them in latent space?** The regret-driven adversarial framework is designed to create a dynamic curriculum where the Cooperator must continually adapt to novel challenges. GOAT identifies underperforming coordination strategies and generates increasingly complex cooperation challenges, compelling the Cooperator to explore and master new behaviors. This iterative process prevents the Cooperator from stagnating into rigid conventions, a common pitfall in self-play or population-based methods. Figure 5a assesses how GOAT explores the latent space of the generative model over the whole training interval. Each point represents a latent vector collected during training episodes 0 to 1750. The Figure 5a compares GOAT embeddings to the distribution of embeddings of the original GAMMA method (Liang et al., 2024) samples throughout training (red ball). We can see that at the start of training, GOAT samples latent vectors distributed throughout this ball (shown in purple). As training progresses, it chooses a particular region to exploit. After a few iterations, it moves to a new region, presumably because the Cooperator has adapted to that strategy, and it is no longer able to maximize regret. Finally, at the end of the training step in yellow, GOAT has traveled across the latent space to explore particular regions (those with high regret) to find valid strategies that exploit weaknesses in the Cooperator.

**RQ5: Ablation study of Minimax vs. Regret.** In Section 4, we hypothesized that even with a generative model in the loop, the purely adversarial minimax objective (Eq. 3) is misaligned with effectively training agents to cooperate. While the generative model can prevent sabotage behavior, because minimax focuses on worst-case scenarios, we hypothesized that a minimax adversary would still focus too much on poorly performing or incompetent partners. In this experiment, we directly compare training the GOAT adversary with either minimax (Eq. 3) or regret (Eq. 4) on the Multi-Strategy Counter Overcooked layout. We further investigate the embeddings chosen by both types of adversary throughout the training period in Figure 5b. We find that the minimax adversary mostly fixates on a single region (red cluster in the plot) where the policies result in low-reward or unproductive behaviors. One such example, shown as a video on our website, is where the minimax agent avoids working with its partner and converges to doing just enough not to sabotage the task. The team still receives rewards as the partner policy keeps completing the task uninterrupted. In contrast, we see that the regret objective causes GOAT to move to multiple modes in the embedding space over the course of training, exploring the latent space more effectively and thus covering a wider range of interesting behaviors. As shown in game videos on our website, it starts by doing one task, then shifts to another task, and then does both tasks simultaneously. We can also see robust training instances like intentionally coming in the way or taking roles.

## 7 CONCLUSION

Our work demonstrates that combining generative modeling with adversarial training provides an effective approach for training AI agents that can coordinate with diverse human partners. Using a generative model and regret-based optimization to constrain the adversary enables GOAT to generate valid, meaningful cooperation partners while posing a curriculum of increasingly challenging scenarios, adapted to the weaknesses of the Cooperator's policy. Thus, we enable robust learning without the risk of degenerate strategies. Experimental results on three cooperation tasks show that GOAT achieves state-of-the-art performance in both simulated evaluations and studies with real human partners. The improved coverage of the partner space demonstrated by our agents suggests this approach can successfully and robustly achieve generalization to human behavior. Future work could explore extending this framework to more complex collaborative tasks and investigating ways to incorporate explicit human feedback during training.

## 8 ACKNOWLEDGMENTS

This research was supported by the Cooperative AI Foundation, the UW-Amazon Science Gift Hub, Sony Research Award, UW-Tsukuba Amazon NVIDIA Cross Pacific AI Initiative (XPAI), the Microsoft Accelerate Foundation Models Research Program, Character.AI, DoorDash, and the Schmidt AI2050 Fellows program. This material is based upon work supported by the Defense Advanced Research Projects Agency and the Air Force Research Laboratory, contract number(s): FA8650-23-C-7316. Any opinions, findings and conclusions, or recommendations expressed in this material are those of the author(s) and do not necessarily reflect the views of AFRL or DARPA.

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

## A    REPRODUCIBILITY

We have our live interactive demo, extended experimental data, and code on our webpage, link on first page.

## B    IMPLEMENTATION DETAILS

**Generative models.** The VAE model was trained on a dataset containing joint trajectories of two players. This dataset was created by evenly sampling from the pairs in the simulated agent population $\{\pi_1, ..., \pi_N\}$, generating 100k joint trajectories. The dataset was split in a 70/30 format for training and evaluation. The VAE was trained using ELBO loss, and linear scheduling of the KL penalty coefficient $\beta$ was applied to control the KL divergence of the posterior distribution. Automatically, checkpoints of VAE models of different strengths were saved. For CMG, the training data used for VAE was 8 policies generated by CoMeDi on CMG. Due to a smaller amount of data, we used data augmentation to train the VAE.

**Cooperator Agent** The Cooperator agent was trained using PPO (Schulman et al., 2017). To promote exploration, the first 100M steps use reward shaping for dish and soup pick-up. We adapted our codebase on the previous implementation of HSP (Yu et al., 2022) and GAMMA (Liang et al., 2024).

**GOAT** For GOAT policy training, we use the REINFORCE objective with KL divergence penalty to keep the policy constrained in the VAE latent space. This objective requires two separate passes of the environment to gather returns for self-play and cross-play.

## C    HYPERPARAMETERS

GOAT uses a simple policy and was trained using REINFORCE to generate latent vectors. The architecture and hyperparameters are defined below.

| hyperparameter | value |
|---|---|
| network architecture | 4-layer MLP |
| hidden dimension | 128 |
| activation function | ReLU |
| output dimension | $2\times$ z_dim |
| optimizer | Adam |
| learning rate | 0.0005 |
| weight decay | 0.0001 |
| latent dimension (z_dim) | 16 |
| KL coefficient | 5.0 |

Table 1: Hyperparameters for GOAT Generator

The Cooperator agent is trained using MAPPO with all overcooked layouts having similar low-level implementation details like architecture and hyperparameters. Every policy network has the same structure, with a CNN coming after an RNN (we use GRU).

The architecture of the policy model and the generative model are comparable. The representations are transformed into a variational posterior and action reconstruction predictions using an encoder head and a decoder head.

## D    COMPUTATIONAL RESOURCES

Our primary tests were carried out on AMD EPYC 64-Core Processor and NVIDIA L40s/L40 clusters. One Cooperator agent can be trained in roughly a day. The primary tests require roughly 24-36 GPU hours for Overcooked and 1 hour for rest of the environments. We perform some preliminary experiments to determine the best training frameworks and hyperparameters.

| hyperparameter | value |
|---|---|
| CNN kernels | [3, 3], [3, 3], [3, 3] |
| CNN channels | [32, 64, 32] |
| hidden layer size | [64] |
| recurrent layer size | 64 |
| activation function | ReLU |
| weight decay | 0 |
| environment steps | 100M (simulated data) or 150M (human data) |
| parallel environments | 200 |
| episode length | 400 |
| PPO batch size | $2 \times 200 \times 400$ |
| PPO epoch | 15 |
| PPO learning rate | 0.0005 |
| Generalized Advantage Estimator (GAE) $\lambda$ | 0.95 |
| discounting factor $\gamma$ | 0.99 |

Table 2: Policy hyperparameters

| hyperparameter | value |
|---|---|
| CNN kernels | [3, 3], [3, 3], [3, 3] |
| CNN channels | [32, 64, 32] |
| hidden layer size | [256] |
| recurrent layer size | 256 |
| activation function | ReLU |
| weight decay | 0.0001 |
| parallel environments | 200 |
| episode length | 400 |
| epoch | 500 |
| chunk length | 100 |
| learning rate | 0.0005 |
| KL penalty coefficient $\beta$ | $0 \rightarrow 1$ |
| latent variable dimension | 16 |

Table 3: Hyperparameters for VAE models

# E    COOPERATIVE REACHING GAME: GAME LAYOUT & 11 HEURISTIC AGENTS

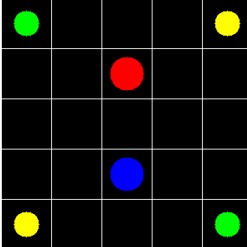

Figure 6: Cooperative Reaching Game (CRG): Red and Blue are the two agents that should cooperate to reach and stay on one of the goal coordinates in the corner. Yellow goal coordinates lead to a reward of 1, and Green goal coordinates lead to a reward of 0.75.

- **Heuristic H01:** Selects actions moving teammates toward the nearest reward-providing coordinate.
- **Heuristic H02:** Selects actions moving teammates toward the reward-providing coordinate most distant from their episode starting position.
- **Heuristic H03:** Moves teammates toward the nearest optimal reward-providing coordinate.
- **Heuristic H04:** Moves agents toward the optimal reward-providing coordinate most distant from teammates' initial episode location.
- **Heuristic H05:** Same as H4, but considers only suboptimal reward-providing coordinates instead of optimal ones.

- **Heuristic H06:** Same as H5, but teammates move toward the nearest suboptimal reward-providing coordinate.
- **Heuristic H07:** At episode start, agents randomly select a reward-providing coordinate and move toward it.
- **Heuristic H08:** Moves teammates toward the reward-providing coordinate nearest to their counterpart agent's location.
- **Heuristic H09:** Same as H8, but considers only optimal reward-providing coordinates as destinations.
- **Heuristic H10:** Moves teammates directly toward their counterpart agent's location.
- **Heuristic H11:** Always randomly selects actions from teammates' available action set.

# F    TESTS ON CMG-S

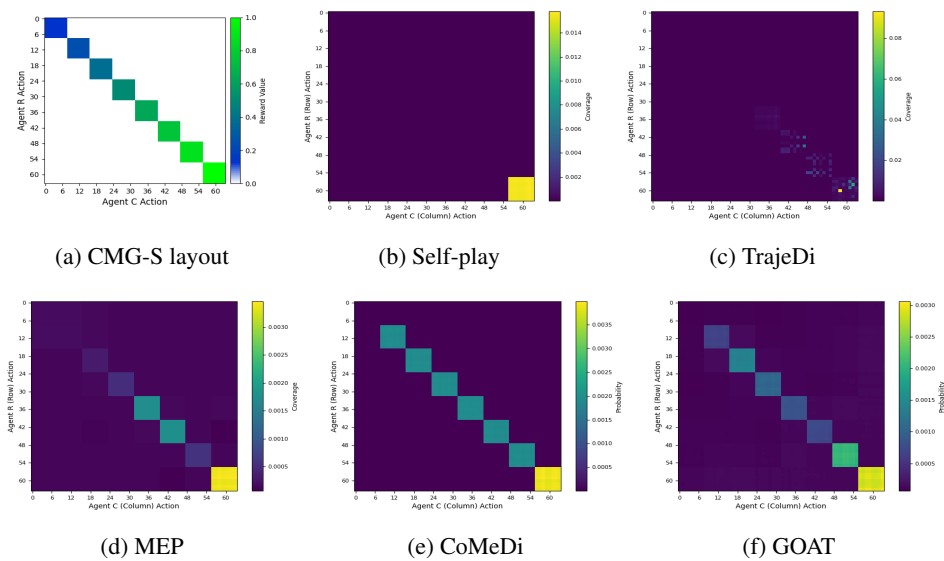

Figure 7: We provide additional tests of the CMG-S problem space. In the problem space, rewards flow in a direction that creates a smooth learning landscape for the algorithms to exploit. The policies for the methods were aggregated and normalized into a heatmap. GOAT successfully assigns priorities to regions.

# G    MINIMAX AND REGRET COMPARISON ON OVERCOOKED

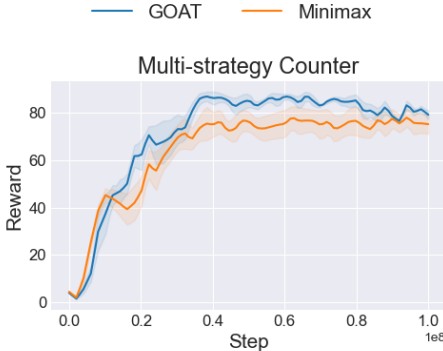

Figure 8: Performance comparison between minimax vs GOAT objective on Multi-Strategy Counter when evaluated against a held-out human behavior cloned model as a partner. Error bars are the Standard Error of the Mean. GOAT outperforms MiniMax, but here MiniMax produces poor-performing agents that fail to participate in coordination. See game videos on our website.

## H   MODE DISCOVERY WITH RESPECT TO POPULATION SIZE.

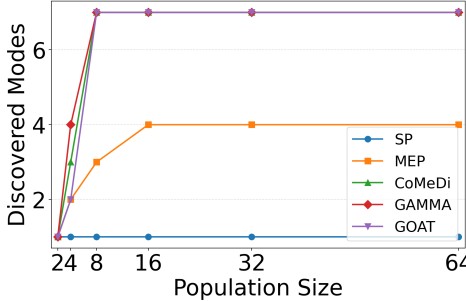

Figure 9: Number of modes discovered by each baseline in Cooperative Matrix Game (CMG) with respect to the population size. CoMeDi, GAMMA, and GOAT discover all the modes with a population size of 8, but each one receives different rewards as shown earlier in Figure 2.

## I   GOAT USING DIFFERENT VAES TRAINED ON DIFFERENT AGENT DATA

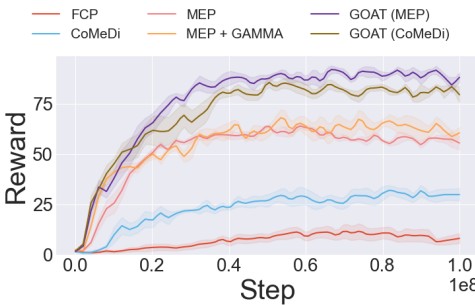

Figure 10: Performance comparison between GOAT using VAE trained with MEP agents data (purple line) vs GOAT using VAE trained with CoMeDi agents data (brown line) on Counter Circuit Overcooked Layout.

## J   COOPERATIVE STRATEGIES LEARNED BY VAE

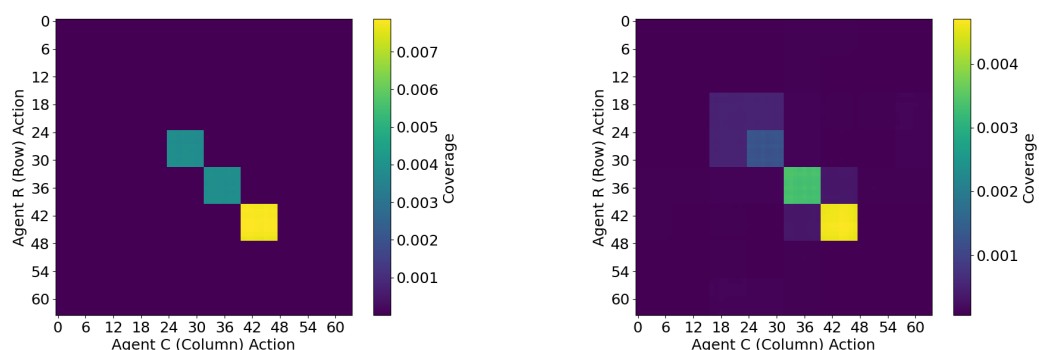

Figure 11: Left: We collect 4 cooperative policies on CMG-M using CoMeDi and then aggregate them. Right: The policies are used for training the VAE, where the latent dimension is 2 and we estimate the policy reconstruction from the VAE.

## K   BROADER IMPACT STATEMENT

Our work on human-AI coordination aims to provide humanity with better cooperative AI that can adapt, understand, and generalize to diverse humans. While this could enhance collaborative robots,

autonomous vehicles, and assistive technologies, our method specifically improves generalization to new partners without requiring extensive human data. GOAT has the potential to improve capabilities for household robots, vehicles, and collaborative AI systems where seamless human interaction adds value to people's lives. This type of technology enhances human modeling, which can be deployed widely to help people from all walks of life and of all age groups. We would also like to note that the development of such an AI system will accelerate technological progress and requires responsible development and deployment.

## L    LIMITATION

The effectiveness of this method depends on the training of VAE models and enough data available for VAE to learn the distribution, instead of overfitting to a few examples. It would be a promising method for fields like robotics and human-AI interaction, where data availability is not an issue. We also found that in sparse reward problems, this method discovers solutions and trains the cooperator robustly.

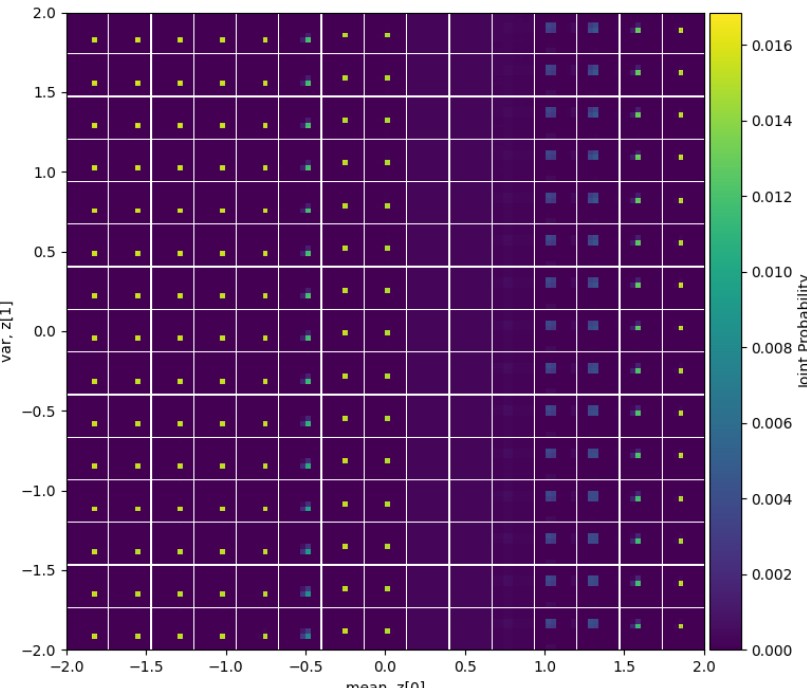

Figure 12: After training a VAE on a 4-population pool of CoMeDi agents, we visualize the generation by interpolating the dimensions of the latent vector. Particularly for this example, we choose the latent dimension to be 2. We can see that VAE trained on cooperative policies has only encoded those policies in the latent space. As we move from one side of the plot to the other, we can see that different cells are activated, and it covers the cooperative policies as seen in the Figure 11

