# OpenReview forum: "Improving Human-AI Coordination through Online Adversarial Training and Generative Models"
_ICLR.cc/2026/Conference — ICLR 2026 Poster_

### Official Review · Reviewer_HUTe · 2025-10-25

**Soundness:** 3
**Presentation:** 4
**Contribution:** 3
**Rating:** 8
**Confidence:** 4

**Summary:**

This paper focuses on the problem of training a cooperative agent in a two player cooperative Markov game. It combines a generative procedure for sampling cooperative partner agents with an adversarial procedure for finding agents whose strategies challenge a learning agent to train an agent that is robust to diverse partner strategies while playing cooperatively and maximizing returns.

The generative procedure uses a VAE similar to previous work. The adversarial procedure works by finding embedding values that would lead to partner agents being sampled by the generator that maximize regret. This regret is calculated by calculating the difference between self-play scores and cross-play scores with the learning agent.

Evaluation is done on a toy matrix game, a reaching environment, and two overcooked environments. Evaluation with human participants in overcooked show that the proposed technique, GOAT, outperforms evaluated baselines in terms of dealing with human strategies.

**Strengths:**

* This paper combines ideas from previous works in a clear and fairly principled manner.
* The idea is explained well. Figure 1 is a good overview of the technique. Section 3 is succinct but clear, and the figures referring to the experiments are also clear and fairly well explained. Some caveats here will be expounded on the weaknesses section
* Human evaluations give the idea a lot of credence. The idea seems to be performing as advertised in the overcooked domain.
* The human evaluation procedure is well designed.
* Appreciate the error bars and their explanation in Figure 4.
* As far as I can tell, all required relevant work seems to be cited.

**Weaknesses:**

* Figures 5 and 6 are a little hard to follow. After reading the text in the experiments and intuiting the intention, I am able to guess what the authors are trying to show. Figure 6 can be understood well in this manner. But Figure 5 was still a little confusing due to the two different dots for GAMMA and GOAT in addition to the gradient colored dots for the episode numbers. If Figure 5 can be simplified it would get the point across much better.
* Nit about how the value function is defined at the end of the first paragraph of Section 3: Value functions are generally conditioned on the state. Perhaps better to phrase this expression as expected returns?
* Figure 3 was also a little difficult to look at. But I appreciate the summation across methods in the last row. Perhaps this row can be separated or highlighted so a reader can look at the summary first before going through the detailed comparison for each heuristic agent. The description would also benefit from mentioning that the maximum sum would be `11`.


Minor: Line 83, the citation should be a `\citep`

**Questions:**

* In the paragraph preceding equation 4, the paper claims that self-play is a valid proxy for optimal score without actually stating that this is a proxy. Would self play actually be optimal in all scenarios? I can imagine scenarios where the two agents need very different policies in order to act optimally. E.g. being when agents need to fulfill different roles. Perhaps this paragraph needs to be phrased more carefully.

---

> ### Author Response · Authors · 2025-11-27
>
> We thank the reviewer for their thoughtful evaluation and constructive feedback. We appreciate the recognition of our paper’s clarity, principled approach, and the value of the human evaluations. We address the feedback below:
>
> ### Figure 3, 5 & 6
> We appreciate the suggestions and concerns about the figures. Below, we are addressing the feedback to improve readability.
>
> - Figure 3:  We have highlighted the “summation across the methods” row and have updated the description to make it easier to show results.
>
> - Figures 5 & 6:  We have simplified the plots to show the spread for both GAMMA and GOAT, along with a succinct explanation of the plots in the caption.
>
> ### Value Function in Section 3
> Thank you for pointing out concerns with the notations. We have updated the notation in the paper, and it is phrased as expected returns. (Line 172)
>
> ### Regarding Self-Play as a valid proxy for optimal score
> We appreciate raising this important conceptual point. We will revise the paragraph to clarify and ensure the text accurately reflects the intended claim. The VAE model was trained on data collected from a population of agents trained using self-play methods to maximize returns. For any particular instance in the game, the self-play policy scores are the best cooperation scores that a policy could typically achieve.
>
> ### Minor Citation Issue
> Thank you for pointing this out. We have corrected the citation issue in the updated preprint.

---

> > ### Comment · Reviewer_HUTe · 2025-11-27
> > **Response to Author Comment**
> >
> > I thank the authors for responding and including my suggestions into their revision.

---

### Official Review · Reviewer_hn3Y · 2025-10-29

**Soundness:** 4
**Presentation:** 4
**Contribution:** 4
**Rating:** 8
**Confidence:** 4

**Summary:**

The authors present GOAT, a method that generates adversarial agents to train a cooperator agent for Human-AI Coordination. The method first utilizes a GAMMA-style VAE to encode an agent population in latent space. Next, the authors propose to train an Adversary to transform the GAMMA latents to Adversarial embeddings, which is then used by the GAMMA decoder to simulate an adversarial agent. The Cooperator Agent and Adversary are jointly trained together in a minimax objective where the adversary tries to maximize the regret (SP to XP gap) between the adversary and Cooperator agent. The authors then evaluate GOAT on 3 cooperative environments, a matrix game, a cooperative reaching game and Overcooked-AI.

**Strengths:**

- The authors provide a intuitive solution to a long-standing problem in the Human-AI/ZSC community, that of generating viable adversarial cooperative agents without sabotaging effects.
- The paper is very well written and structured.

**Weaknesses:**

- The authors cites two recent/concurrent related works [1] and [2] but did not compare GOAT with the two methods in the experimental sections. It will be interesting to see how GOAT compares to the two methods especially ROTATE as they also proposes a regret like objective to generate a curriculum of adversarial agents to train a cooperator agent.
- GOAT is evaluated on fairly simple, fully observable grid-world environments which limits its generalizability to more complex environments such as those with partial observable states [3] or continuous actions [4].

[1] Wang, C., Rahman, A., Cui, J., Sung, Y., & Stone, P.  ROTATE: Regret-driven Open-ended Training for Ad Hoc Teamwork. arXiv preprint arXiv:2505.23686.

[2] Villin, V., Buening, T. K., & Dimitrakakis, C.  A Minimax Approach to Ad Hoc Teamwork. In Proceedings of the 24th International Conference on Autonomous Agents and Multiagent Systems (pp. 2105-2114).

[3] Gessler, T., Dizdarevic, T., Calinescu, A., Ellis, B., Lupu, A., & Foerster, J. N. OvercookedV2: Rethinking Overcooked for Zero-Shot Coordination. In The Thirteenth International Conference on Learning Representations.

[4] Kang, X., Lee, S. W., Liu, H., Wang, Y., & Kuo, Y. L.  Moving Out: Physically-grounded Human-AI Collaboration. arXiv preprint arXiv:2507.18623.

**Questions:**

- In Figure 5, the authors show that the adversarial latent vectors sampled by GOAT is significantly far away from the cluster of GAMMA latent vectors. As I understand, the GAMMA latent vectors can be interpreted as an interpolation between actual agent trajectories encoded in latent space during the VAE training process. Does this imply that GAMMA could extrapolate new agents out of the standard trained distribution?
- Why is REINFORCE chosen over PPO to train the Adversary?
- The authors state that the main limitation of GOAT is the reliance on a trained VAE model. Could the authors comment on what might the minimally viable population size/amount of data to train a viable VAE model for GOAT?
- Relatedly, does the type of agents (CoMeDi vs MEP) matter when it comes to VAE training?  Does it affect the quality of latent vectors of the VAE?

---

> ### Author Response · Authors · 2025-11-27
>
> We thank the reviewer for the positive assessment of the paper’s contribution, clarity, and soundness. Below, we address each concern concisely.
>
> ### Related work [1] ROTATE & [2] Minimax-Bayes:
> We thank the reviewer for pointing out these recent works. Both ROTATE and Minimax-Bayes represent some similarities with GOAT. In the case of the Minimax-Bayes approach, the method is closely related to the minimax vs regret ablation studies we presented in the experimental section. ROTATE and GOAT share the same high-level intuition of using regret, but ROTATE introduces several constraints that make a direct comparison difficult. Some of the design choices mentioned in ROTATE could significantly increase the computation requirements, which may not result in a fair comparison. Moreover, results posted in ROTATE seem comparable to CoMeDi, which we already include as a baseline in our evaluation.
>
> ### Evaluating GOAT:
> We acknowledge the concern about the environmental complexity for testing GOAT. We selected the following testbeds because they were used by many prior works [3,4,5,6,7] in this line of research. Real-time human evaluation is essential in ZSC and Human-AI coordination because humans do not behave like synthetically generated agents. Overcooked platform, in particular, is still considered a state-of-the-art real-time human-AI evaluation.
>
> ### Interpretation of Figure 5:
> Thank you for this important question that differentiates GOAT from GAMMA.
> The VAE model is trained with different KL regularization coefficients, thereby aligning the latent space differently. With stronger KL regularization, we observe that the latent space is more aligned with the standard normal distribution, reducing variability and limiting disentanglement. With relaxed KL regularization, the latent space is allowed to spread out and encode meaningful structure of the distribution, which also provides variability and disentanglement.
>
> In GAMMA, the best results were achieved with higher KL regularization, which is closer to the standard normal distribution. In contrast, for GOAT, the KL regularization was balanced. GAMMA uses the standard normal distribution to sample latent vectors throughout training, whereas GOAT samples latent vectors influenced by the regret.
>
> ### Choice of REINFORCE vs PPO:
> We appreciate the question on the choice of training algorithm for the adversary. The adversary in GOAT is a one-step policy that chooses latent vectors at the start of each episode. While both algorithms should work for our method, we found REINFORCE simpler and more appropriate. Since adversary policy requires a single-step operation, we do not gain anything from the PPO’s discounted returns or value function estimation. REINFORCE has fewer hyperparameters, no critic, no clipping parameters, and requires no discounting. More importantly, the adversary must be able to jump from one region of the latent space to another. PPO’s clipped objective resists this jump and requires careful tuning of clipping ranges and learning rate schedules.
>
> ### Minimum Viable Population Size for VAE Training:
> A population size of just 8 was sufficient to train GOAT's VAE, compared to the 16–64 typically required by other methods [3], [4], and [7]. GOAT performs better as long as the data coverage for any given task is adequate to generate diverse agents. Please refer to Section H, where we show that a population size of 8 reasonably covers all the modes in the Cooperative Matrix Game from Figure 2. As long as the coverage of the task space is reasonable, GOAT can utilize it to simulate diverse scenarios during training and train the cooperator agent robustly.
>
> ### (CoMeDi vs MEP) for VAE Quality:
> We will add additional experiments on this by using two different VAEs, one trained with MEP [4] and another with CoMeDi [5]. We will compare the performance of both the VAEs with GOAT on one of the Overcooked layouts.
>
>
> [1] Wang, C., Rahman, A., Cui, J., Sung, Y., & Stone, P. ROTATE: Regret-driven Open-ended Training for Ad Hoc Teamwork. arXiv preprint arXiv:2505.23686.
> [2] Villin, V., Buening, T. K., & Dimitrakakis, C. A Minimax Approach to Ad Hoc Teamwork. In Proceedings of the 24th International Conference on Autonomous Agents and Multiagent Systems.
> [3] Strouse et al. Collaborating with Humans without Human Data. NeurIPS 2021.
> [4] Rui Zhao, Jinming Song, Yufeng Yuan, Hu Haifeng, Yang Gao, Yi Wu, Zhongqian Sun, and Yang Wei. Maximum entropy population-based training for zero-shot human-ai coordination, 2022.
> [5] Sarkar, Bidipta, Andy Shih, and Dorsa Sadigh. "Diverse conventions for human-AI collaboration." Advances in neural information processing systems 36 (2023)
> [6] Yancheng Liang, Daphne Chen, Abhishek Gupta, Simon S. Du, and Natasha Jaques. Learning to cooperate with humans using generative agents, 2024
> [7] Rujikorn Charakorn, Poramate Manoonpong, and Nat Dilokthanakul. Generating diverse cooperative agents by learning incompatible policies 2023

---

> > ### Author Response · Authors · 2025-11-28
> >
> > ### Update: (CoMeDi vs MEP) for VAE Quality:
> > We would like to kindly direct the reviewer to Figure 10 in Section I of the updated revision, where we demonstrate that GOAT performs comparably across different VAE models trained with different agent data on counter circuit Overcooked Layout.

---

### Official Review · Reviewer_Zoti · 2025-11-02

**Soundness:** 3
**Presentation:** 4
**Contribution:** 3
**Rating:** 6
**Confidence:** 3

**Summary:**

The paper tackles the problem of learning cooperative agents that are robust to a diverse distribution of human and artificial agents. The models uses a pre-trained generative model to simulate valid cooperative agent policies with adversarial training to maximize regret. The policy itself tries to minimize regret, this induces an adversarial setup, but the challenge is to not the adversary be adversarial _and_ cooperative - two conflicting objectives. If the adversary is not constrained to such policies, then it can become overly combative, resorting to sabotage instead of making the policy robust and foolproof. In contrast to zero-sum games like chess where self-play can improve the performance of the policy by exploiting weaknesses against itself, in cooperative games there is no such incentive for a policy.
To prevent cross-play performance from self-sabotaging, the model essentially samples adversarial players from a learned distribution of agents using a variational autoencoder. this limits the capability for self-sabotage since the agents are sampled from a predefined distribution. This makes the contribution of the work relevant to the community.

**Strengths:**

**Strengths**
1. The VAE model and generative policy allows for sampling policies parametrically with a smooth latent space, preventing expensive zeroth-order methods to sample policies. This makes the optimization objective more tractable and inexpensive. This idea is also used in controllable image generation and editing where latent variables are optimized to achieve a test-time objective with a pretrained network.
2. The formulation is pretty straightforward - given a cooperator policy $\pi_C$, the adversarial policy chooses a partner policy $\pi_P$ to minimize the value function with cross play of the cooperator and partner, while maximize the self-play performance of the partner policy - this is presumably to prevent selecting incompetent policies. This is also interpreted (accurately) as the regret of choosing a cooperative policy over itself in a cross-play scenario. Regret for bad partner policies will be lower (due to low first term in Equation (4)) than a good partner policy that does not work well with the cooperator policy.
3. The experiment setup is satisfactory - including a cooperative matrix game allows easy visualization of one-step policies and analytically verifying optimal adversarial policies for a given cooperative policy, a 5x5 cooperative reaching game with more complexity but still tractable, and a hard game with a larger action space (Overcooked) with real-time dynamics on which performance is also shown with human cooperative players.
4. Extensive analysis with related methods - GAMMA and MinMax is shown in RQ4 and RQ5. Figure 5 shows the adversarial objective drifting to new policies that significantly deviate from the normal distribution (in the projection space), showing that the model is indeed aiming to find harder partner policies.

**Weaknesses:**

1. The robustness of the policy depends on the coverage / support of the autoencoder and its simulated policies. The paper uses fixed agent populations with training methods (Line 178).
2. The paper is not self-sufficient in terms of mentioning how the distribution of partner policies are learned before the adversarial sampling is performed.
3. Figures 5 and 6 are nice, but they also show that the policies deviate a lot from random normal as training progresses. Since the generative model works off a VAE, are the policies that have a latent so far from the normal distribution meaningful at all? Is a regularization apart from the self-play regret (as mentioned in Line 722) enough to constrain the space of sampled policies?
4. Figures 5 and 6 also show that the coverage of partner policies is very concentrated and lacks diversity - could it lead to a possible oscillation of the cooperative policy if certain partner policies are not compatible with each other?

Minor nits:
1. Line 75: "The regret objective proves effective because it challenges the learning agent with a curriculum of increasingly difficult, yet still feasible tasks." - Regret minimization by itself does not motivate a curriculum learning approach. This line could say something like "it challenges the learning agent and expands the frontier / coverage of harder yet feasible state configurations"

**Questions:**

**Questions**

1. Since the distribution of the partner policies is learned using a VAE, why cant the cooperative agent be trained to minimize regret over the entire distribution of partner policies (by sampling partner policies from the VAE) instead of using an adversarial method to sample partner policies? Is that going to be slower or faster than the proposed training in terms of environment interactions and training time?
2. Line 376 - "We hypothesize this is because the generative model, trained on cooperative trajectories, encodes a latent space rich in strategic variations." - the paper posits that the generative model encodes a rich space of strategic variations is what leads to fast convergence - this approach is also used in other prior work (some of which are mentioned in the paper). Does GOAT perform better than prior work leveraging generative policies due to its adversarial sampling?

Minor questions:
1. Line 45/74: Why is Adversary capitalized?
2. uncommon notation for distribution, i.e. $\Delta$ is used. This could be replaced with a better notation in my opinion
3. Line 363: "PBT methods rely on simulated populations of self-play partners, are often computationally expensive, and have poor coverage of actual human behaviors."  - why is cross play and adversarial training less expensive than self play? I thought the tradeoff was to spend more compute on adversarial training to have better mode coverage of adversarial policies. Figures 4b and 4e show that GOAT is indeed faster but is there any argument or justification as to why that is the case, especially because adversarial training can be slow and brittle.

---

> ### Author Response · Authors · 2025-11-27
>
> We appreciate the reviewer’s accurate summary of the paper and acknowledgment of the relevance and contribution of the work. Below, we clarify the raised issues and provide additional clarification where needed
>
> ### Coverage/support of autoencoder with fixed agent population:
> We are comparing the method against other PBT methods, and to be fair in evaluations, we used a fixed agent population of 8 agents across all methods. We chose an 8-agent population because that was reasonable for a good coverage of the task. Please refer to Section H, where we show that a population size of 8 covers all the modes in the Cooperative Matrix Game from Figure 2. As long as the coverage of the task space is reasonable and can be generated by VAE, GOAT can utilize it to simulate diverse scenarios in training.
>
> ### Figures 5 and 6 policies deviate a lot from the random normal:
> The VAE model trained with different KL regularization coefficients $\beta$ aligns the latent space in different ways. With stronger KL regularization, we observe that the latent space is more aligned with the standard normal distribution, reducing variability and limiting disentanglement. With relaxed KL regularization, the latent space is allowed to spread out and encode meaningful structure of the distribution, which also provides variability and disentanglement. For GAMMA, the best results were achieved with higher KL regularization, whereas for GOAT, the KL regularization was balanced. Additionally, we also provide gameplay videos of simulated partner policies during training on our website.
>
> ### Regret and Curriculum Interpretation:
> Thank you for pointing out inconsistencies in the text. We have addressed this in the paper as “The regret objective proves effective because it challenges the learning agent by continuously identifying difficult but achievable scenarios where the agent currently underperforms.”
>
> ### How partner distributions are learned:
> We acknowledge the need to specify the VAE training scheme and have addressed this point by adding the description of how partner strategies are learned using VAE in Section 3 (Line 175-192).
>
> ### Questions:
>
> ### Why can't the cooperative agent be trained to minimize regret over the entire distribution of partner policies (by sampling partner policies from the VAE) instead of using an adversarial method to sample partner policies?
> The method proposed by the reviewer would be similar to GAMMA [1] with a regret-based objective. Just sampling partners from the VAE is much less efficient at identifying the weaknesses of the learning agent. GOAT purposely searches for latent vectors that challenge the learning agent, thus becoming sample-efficient. Robustly training the learning agent requires frequent training on the rare challenging scenarios, where uniform random sampling falls short.
>
> ###  Does GOAT perform better than prior work leveraging generative policies due to its adversarial sampling?
> Yes, GOAT does perform better than prior work that leverages generative policies such as GAMMA [1]. We present the comparison in Figures 2, 3 & 4 for different tasks.
>
> ### Why is adversary capitalized?
> We capitalize “Adversary” and “Cooperator” to emphasize that they are distinct agents with distinct roles in our game formulation. We are happy to use lowercase if that is preferred and makes the concept clearer.
>
> ### Cross-Play and Adversarial Training vs PBT
> This is a valid point, and we will rephrase the claim. GOAT achieves faster convergence in Figures 4b and 4e because adversarial exploration of the generative model’s latent space enables GOAT to rapidly identify latent vectors that exploit weaknesses in the Cooperator's policy.
>
> ### Notation for distribution
> We have addressed this in Section 3 in the updated paper.
>
> [1] Yancheng Liang, Daphne Chen, Abhishek Gupta, Simon S. Du, and Natasha Jaques. Learning to cooperate with humans using generative agents, 2024

---

### Meta-Review · Area_Chair_pzwz · 2026-01-09

**Summary:**

Across three reviews, the main concerns were: (i) positioning and missing comparisons to very recent/concurrent regret/minimax approaches (notably ROTATE and a minimax-Bayes style method) and how GOAT differs empirically and conceptually; (ii) generality and evaluation scope, since experiments are primarily in relatively simple, fully-observable domains (plus Overcooked), leaving open how well the approach transfers to more complex settings (e.g., partial observability or continuous control); (iii) methodological clarity/assumptions, including whether the adversarial search may push embeddings far from the VAE’s nominal latent distribution, how much robustness depends on the support/coverage of the pretrained partner-policy distribution, and whether the paper was sufficiently self-contained in describing how partner policies are learned; and (iv) presentation/notation issues, including figure readability, distribution notation, and carefully phrasing “self-play as proxy for optimal” (role-asymmetry caveat).

Despite these limitations, the reviewers agreed the paper offers a clear, principled synthesis (generative partner modeling + regret-driven adversarial search) and, importantly, includes human evaluation in Overcooked, which substantially strengthens the evidence for the intended Human–AI coordination use case.

**Reviewer Concerns:**

Addressed concerns

- Figure readability (Figures 3/5/6): Authors simplified Figures 5/6, added clearer captions, and highlighted the “summation” row in Figure 3 to improve readability.

- Notation and value-function phrasing: Authors updated notation (including distribution notation) and rephrased the “value function” point as expected returns to avoid confusion.

- “Self-play as proxy for optimal” phrasing: Authors agreed the original phrasing could overclaim; they revised text to clarify the intended meaning and contextualize it with how the VAE training data was collected (self-play-trained populations).

- Cross-play/adversarial training vs PBT claim: Authors agreed the claim needed rewording; they revised it and justified the observed faster convergence as coming from adversarial latent-space exploration that quickly discovers challenging partners.

- Latent-space extrapolation / interpretation of far-from-normal latents (Figure 5): Authors provided an explanation grounded in KL-regularization choices during VAE training and emphasized GOAT’s sampling is “regret-influenced” rather than standard-normal sampling; they clarified why this does not necessarily imply meaningless partners, within their framework.

- Algorithm choice for training the Adversary (REINFORCE vs PPO): Authors argued REINFORCE is sufficient and simpler for a single-step latent selection policy and avoids PPO’s clipping/tuning complications when the adversary needs to “jump” across latent regions.

- Minimum viable population size for VAE training: Authors stated that a population size on the order of 8 was sufficient in their tests (with the caveat that adequate mode coverage in the task distribution remains important).

Outstanding concerns

- Missing head-to-head comparisons to concurrent methods (ROTATE; minimax-Bayes style): Authors discussed why comparisons may be non-trivial (different constraints/compute assumptions; overlap with included baselines), but did not provide direct experimental comparisons. This remains a substantive gap.

- Generality beyond current benchmarks (complexity/partial observability/continuous control): Authors justified their benchmark choices as standard in Human–AI coordination literature and emphasized human evaluation in Overcooked; however, the broader generalization claims remain only partially supported empirically.

- Dependence on partner-distribution support/coverage: The concern that robustness hinges on the learned partner-policy distribution’s coverage remains inherently true; the rebuttal clarifies but does not fully resolve this limitation with additional evidence or diagnostics.

- Out-of-distribution latent validity: While the rebuttal provides a plausible explanation (KL regularization + regret-driven search), the question of whether very far-off latents always decode to semantically meaningful, cooperatively-intended policies is not fully settled by additional quantitative checks.

**Reviewer Scores:**

- Reviewer Zoti: Likely no change (6). The rebuttal addressed notation/clarity and provided explanations, but the reviewer’s core reservations (coverage dependence; latent validity; broader generalization) largely remain.

- Reviewer hn3Y: Likely no change (8). Their key concern—missing comparisons to concurrent works—was acknowledged but not fully resolved; nevertheless they were already positive and the rebuttal addressed most technical questions.

- Reviewer HUTe: Likely no change (8). Their main issues were presentation/clarity, and they explicitly acknowledged the revision incorporated their suggestions.

---

### Decision · Program_Chairs · 2026-01-26

Accept (Poster)